# Association of the rs9896052 Polymorphism Upstream of *GRB2* with Proliferative Diabetic Retinopathy in Patients with Less than 10 Years of Diabetes

**DOI:** 10.3390/ijms251910232

**Published:** 2024-09-24

**Authors:** Caroline Moura Cardoso Bastos, Lucas Marcelo da Silva Machado, Daisy Crispim, Luís Henrique Canani, Kátia Gonçalves dos Santos

**Affiliations:** 1Laboratory of Human Molecular Genetics, Lutheran University of Brazil (ULBRA), Av. Farroupilha 8001, Canoas 92425-900, RS, Brazil; carolmoura98@gmail.com (C.M.C.B.); lucmar1997@gmail.com (L.M.d.S.M.); 2Endocrine Division, Clinical Hospital of Porto Alegre (HCPA), R. Ramiro Barcelos 2350, Porto Alegre 90035-903, RS, Brazil; dcmoreira@hcpa.edu.br; 3Department of Internal Medicine, Federal University of Rio Grande do Sul (UFRGS), R. Ramiro Barcelos 2400, Porto Alegre 90035-003, RS, Brazil; lcanani@hcpa.edu.br

**Keywords:** type 2 diabetes mellitus, diabetic retinopathy, GRB2, polymorphism, rs9896052, genetic association

## Abstract

Growth factor receptor-bound protein 2 (GRB2) is a negative regulator of insulin signaling and a positive regulator of angiogenesis. Its expression is increased in a mouse model of retinal neovascularization and in patients with type 2 diabetes mellitus (T2DM). This case–control study aimed to investigate the association between the rs9896052 polymorphism (A>C) upstream of *GRB2* and proliferative diabetic retinopathy (PDR) in patients with T2DM from Southern Brazil, taking into consideration self-reported skin color (white or non-white) and the known duration of diabetes (<10 years or ≥10 years). Genotypes were determined by real-time PCR in 838 patients with T2DM (284 cases with PDR and 554 controls without DR). In the total study group and in the analysis stratified by skin color, the genotype and allele frequencies were similar between cases and controls. However, among patients with less than 10 years of diabetes, the C allele was more frequent in cases than in controls (63.3% versus 51.8%, *p* = 0.032), and the CC genotype was independently associated with an increased risk of PDR (adjusted OR = 2.82, 95% CI 1.17–6.75). In conclusion, our findings support the hypothesis that the rs9896052 polymorphism near *GRB2* is associated with PDR in Brazilian patients with T2DM.

## 1. Introduction

Diabetic retinopathy (DR) is a neurovascular complication of diabetes mellitus (DM) and the leading cause of blindness in working-age adults [1], with a global prevalence of 22% [2]. DR can be classified into non-proliferative (NPDR) and proliferative (PDR) stages, according to its severity. Non-proliferative DR is characterized by a breakdown of the blood–retinal barrier (BRB), a loss of pericytes, microaneurysms, and retinal hemorrhages, among other microvascular abnormalities that may range from mild to severe. As ischemia and hypoxia develop, severe NPDR may progress to PDR, with the hallmark of neovascularization in the retina, which may lead to vitreous hemorrhage, retinal detachment, and vision loss [3,4]. These abnormalities are triggered by long-term hyperglycemia, which damages the neurovascular unit of the retina through direct effects on the endothelium [3] and indirectly through cell signaling pathways that induce the production of proinflammatory cytokines, adhesion molecules, chemokines, reactive oxygen species, and growth factors, thus promoting vascular leakage and pathological angiogenesis. Ultimately, these changes disrupt the normal interaction between the neuronal and vascular components of the retina and impair retinal function [3,4,5].

In addition to prolonged DM, chronic hyperglycemia, hypertension, and dyslipidemia are the main risk factors for the onset and progression of DR [1]. However, these factors explain at most 10% of cases of progressive DR and incident PDR [6]. Although DR is recognized as a multifactorial disorder resulting from the interplay between genetic, epigenetic, and environmental factors [7,8], the gene variants underlying this complication remain elusive [7,9]. In this scenario, a genome-wide association study (GWAS) conducted by Burdon et al. [10] identified novel risk loci for sight-threatening DR (defined as severe NPDR, PDR, or macular edema) in white Australians with type 2 DM (T2DM), including a genetic variant located 17 kb upstream of the growth factor receptor-bound protein 2 (*GRB2*) gene on chromosome 17q25.1 (rs9896052; A>C). The initial signal of association between the rs9896052 polymorphism and sight-threatening DR was the only one that was replicated in three other cohorts in the same study [10].

GRB2 is an adaptor protein that connects activated receptor tyrosine kinases with downstream effectors in the signal transduction and cell communication pathways mediated by phosphatidylinositol 3-kinase (PI3K)/AKT and Ras/mitogen-activated protein kinase (MAPK) signaling. The activation of the PI3K/Akt and Ras/MAPK cascades mediates the activities of growth factors, promoting cell proliferation, differentiation, survival [11,12,13], and angiogenesis [13,14]. GRB2 is essential for early embryonic development [15] but is overexpressed in many types of tumors, where it has an oncogenic effect [14]. In the GWAS by Burdon et al. [10], the minor A allele of the rs9896052 polymorphism was associated with an increased risk of sight-threatening DR in patients of European ancestry with type 1 or type 2 DM and in Indians with T2DM. In addition, the authors showed that the GRB2 protein was expressed in all layers of the normal human and mouse retina, and it was upregulated in a mouse model of retinal stress and neovascularization after selective Müller cell ablation [10]. In this mouse model, a patchy loss of Müller cells leads to photoreceptor degeneration, BRB breakdown, and intraretinal neovascularization, mimicking the major features of DR [16].

Following the report by Burdon et al. [10], we were prompted to investigate whether the rs9896052 polymorphism near *GRB2* is associated with PDR in outpatients with T2DM from Southern Brazil. This potential association was also examined by stratifying the patients into groups according to their self-reported skin color (white or non-white) and the duration of their diabetes (<10 years or ≥10 years).

## 2. Results

### 2.1. Characteristics of Study Population

Subjects with T2DM (*n* = 838) ranged in age from 32 to 88 years (mean age, 60.0 ± 9.5 years), and most were female (55.3%) and white (77.7%). Similarly, most of the blood donors (*n* = 106) were white (78.3%), but slightly more than half were male (52.8%) and younger than the patients with T2DM (mean age, 42.2 ± 8.6 years; range, 30–69 years).

The basic characteristics of the cases (patients with PDR) and controls (patients without DR) are provided in Table 1. Cases were older, more often male, daily insulin users, had diabetes for longer, had higher systolic blood pressure, had a lower body mass index (BMI) and HDL cholesterol, and had more impaired renal function than the controls. In contrast, cases were less often active smokers than controls (Table 1).

### 2.2. Genotype and Allele Frequencies of rs9896052 Polymorphism in the Study Groups

The frequency distribution of the rs9896052 polymorphism in blood donors, the overall group of patients with T2DM, and patients with PDR (cases) and without DR (controls) is shown in Table 2. The genotype frequencies did not deviate from those predicted by the Hardy–Weinberg equation for all groups of subjects. The genotype and allele frequencies were virtually identical in blood donors and in the overall group of patients with T2DM and were similar in cases and controls (Table 2). A logistic regression analysis revealed no association between rs9896052 polymorphism and PDR (Appendix A).

Next, we examined whether patients with T2DM with different rs9896052 genotypes differed in terms of their clinical and demographic characteristics. This analysis revealed that the frequency of the rs9896052 polymorphism varied between white and non-white subjects. Moreover, patients with the CC genotype had lower levels of glycated hemoglobin and a lower estimated glomerular filtration rate (eGFR) than those with the AA genotype (Appendix A), reflecting that white subjects had lower levels of these two clinical biomarkers than non-white subjects. The analysis of the distribution of the rs9896052 polymorphism in the study groups was repeated by stratifying by skin color to assess whether this characteristic could be a confounding or interacting factor in the relationship between the rs9896052 polymorphism and PDR. The C allele was the major allele among white subjects (57%) and the minor allele among non-white subjects (37–38%). Again, the genotype and allele frequencies were similar in blood donors and the overall group of patients with T2DM, and they were not significantly different between cases and controls in both white and non-white subjects (Table 3). No association between the rs9896052 polymorphism and PDR was detected in our logistic regression analysis (Appendix A).

To evaluate the possible interaction between the rs9896052 polymorphism and the duration of diabetes, the analyses were repeated according to this variable (<10 years versus ≥10 years). In the short-duration group, the CC genotype was more frequent in cases than in controls (*p* = 0.016 in the analysis of adjusted residuals), as was the C allele (Table 4). After adjusting for gender, age, skin color, eGFR, glycated hemoglobin, BMI, HDL cholesterol, and LDL cholesterol, the CC genotype remained associated with an increased risk of PDR (Table 5). In those with a longer duration of diabetes, no differences in rs9896052 distribution were observed between cases and controls (Table 4), and no association was identified in the regression logistic analysis (Appendix A).

## 3. Discussion

In this study, we found an association between the CC genotype of the rs9896052 polymorphism near *GRB2* and PDR in Southern Brazilians with a known T2DM duration of less than 10 years. This association was not observed in patients with a longer duration of diabetes. This suggests that genetic variants favor the progression to PDR in those with a short duration of diabetes, that genetic effects are more evident in those with a shorter duration of diabetes, or that comorbidities and other clinical factors overlap or have a greater impact on the development of PDR than genetic susceptibility from the second decade after a diabetes diagnosis.

GRB2 is a negative regulator of insulin signaling in HepG2 [17] and adipocytes in culture [18] and a positive regulator of angiogenesis [19,20]. In a T1DM mouse model, islet transplantation markedly reduced cell apoptosis and promoted the neovascularization of islet grafts because of the increased expression of GRB2 and growth factor-related proteins, including vascular endothelial growth factor A and insulin-like growth factor 1 receptor [20]. Another study showed that GRB2 expression was increased in human retinal pigment epithelial (hRPE) cells exposed to high glucose in a concentration-dependent manner, and its overexpression was accompanied by an increase in the expression of proinflammatory cytokines [21]. Serum GRB2 was found to be positively correlated with C-reactive protein and interleukin-6 levels, which were higher in patients with T2DM (compared to the healthy population), and further increased in patients with T2DM with carotid atherosclerosis [22].

Taken together, the experimental and clinical evidence shows that *GRB2* is a promising susceptibility gene for severe forms of DR. However, the mechanistic link between a genetic variant near *GRB2* and DR remains to be investigated. Moreover, no study on the association between the rs9896052 polymorphism upstream of *GRB2* and DR has been reported after the paper by Burdon et al. [10]. In their study, the A allele was identified as a risk allele for sight-threatening DR in South Indian and white subjects from Australia and the United Kingdom [10]. In contrast, in the present study, the C allele was identified as being associated with an increased risk of PDR in the South Brazilian population. This discrepancy is interesting, but not surprising, as the genuine association of a complex disease with opposite alleles of the same polymorphism in different populations can occur because of differences in the linkage disequilibrium (LD) pattern and interaction of the investigated variant with a causal variant in another locus and environmental factors [23,24]. Causal disease effect sizes and/or causal variants were suggested to be more population-specific in functionally important regions, including conserved and regulatory regions, which could be explained by stronger gene–environment interactions at loci affected by selection [25]. Indeed, the rs9896052 polymorphism is within a 150 kb block of a LD containing several genes that are expressed in the retina and is located between two H3K27Ac epigenetic marks [10].

In our study, the general frequency of the A allele was 47%, similar to that described in white subjects and South Indians with T2DM [10]. However, it was more frequent in non-white subjects (62%) than in white subjects (43%), which is consistent with the frequencies reported in European and African populations (https://www.ncbi.nlm.nih.gov/snp/rs9896052, accessed on 2 July 2024). *GRB2* was upregulated during retinal stress and neovascularization in an animal model, and the rs9896052 polymorphism was independently associated with severe DR in white patients with T1DM and T2DM, as well as in South Indians [10] and South Brazilians with T2DM. However, it is not known whether the rs9896052 polymorphism has an effect (or not) on *GRB2* expression because its potential functionality has not been assessed in any studies.

Despite the differences in allele frequencies between white and non-white subjects in our study, skin color did not seem to have contributed to the lack of association observed between the rs9896052 polymorphism and PDR in the overall sample of patients with T2DM. Interestingly, in the study by Burdon et al. [10], the A allele was associated with sight-threatening DR in a European-descent replication cohort with T2DM only after adjusting for age, sex, the duration of diabetes, hypertension, nephropathy, and glycated hemoglobin, reinforcing that these clinical variables may interact with or confound the association of a genetic variant with DR in a population-specific manner.

In this context, a factor that must be considered is the duration of diabetes. Several authors have highlighted that diabetes’ duration may modify the association between genetic variants and late diabetic complications, and, if ignored, can lead to power loss [26,27,28]. In our study, the CC genotype of the rs9896052 polymorphism was associated with PDR only among patients with a known T2DM duration of less than 10 years. Although patients who had had T2DM for longer had a slightly better lipid profile (with higher HDL and lower triglyceride levels) than those with a shorter duration of diabetes, they were more often regular insulin users, hypertensive (with a higher systolic blood pressure), and had a lower eGFR (Appendix A). Assuming that there is a true association between the rs9896052 polymorphism and PDR in our population, even if of a small magnitude, it would be plausible to expect that the genetic variant might have a stronger effect on the progression of DR in the first years after the diagnosis of diabetes, whereas it no longer has an impact after a longer period of diabetes, in view of the presence of other risk factors for PDR. Another theoretical explanation for the lack of association between genetic risk factors and diabetic complications in patients with more than 10 years of diabetes is survival bias [28]. Patients with the CC genotype could have developed PDR in the first years after their diabetes diagnosis, along with other chronic complications and serious vascular events, making these patients more vulnerable to their consequences, such as death. This could explain why the frequency of the CC genotype was reduced among PDR cases with at least 10 years of diabetes (26.8%) in comparison to the PDR cases with less than 10 years of diabetes (43.3%) in our study.

Our findings should be considered in the context of some limitations related to the data analysis conducted. Stratified analyses by skin color and the known duration of diabetes may have led to the positive association observed between the rs9896052 polymorphism and PDR in patients with less than 10 years of diabetes (a type I error due to multiple testing). This association could have also resulted from the small number of PDR cases with a shorter duration of diabetes (*n* = 60). Nevertheless, our findings were based on a relatively large number of subjects from a genetically diverse population other than those originally investigated in a GWAS searching for genetic determinants of DR, PDR, and diabetic macular edema [9,10].

## 4. Materials and Methods

### 4.1. Ethical Statement

This case–control study was approved by the Institutional Review Board of Universidade Luterana do Brasil (ULBRA) in December 2019, under CAAE number 26567119.8.0000.5349, and consolidated review number 3.774.110. All procedures were performed in compliance with the current guidelines and regulations for human research. Written and signed informed consent was obtained from all study participants before blood and data collection.

### 4.2. Subject Enrollment and Data Collection

The study population comprised 838 outpatients with T2DM and 106 blood bank donors with no personal or first-degree family history of diabetes. Of the 838 patients with T2DM included in this study, 284 had PDR (cases) and 554 did not have DR (controls). DR was classified on the basis of the most severely affected eye. Proliferative DR was defined as new vessels on or near the optic disc and/or neovascularization elsewhere in the retina, with or without vitreous or preretinal hemorrhage. Patients who previously underwent panretinal photocoagulation were included in the case group. Retinopathy was considered to be absent if the patient had no fundus abnormalities [29].

In this study, we used clinical data and DNA samples from patients with T2DM who were enrolled over two different periods (2000–2010 and 2015–2017) and included in previous reports by our research group [30,31]. Patients enrolled from 2000 to 2010 (*n* = 574) regularly attended the outpatient clinic of one of the following public institutions in Rio Grande do Sul (RS) state in Southern Brazil: Hospital de Clínicas de Porto Alegre (HCPA), Grupo Hospitalar Conceição (Porto Alegre/RS), Hospital São Vicente de Paulo (Passo Fundo/RS), and Fundação Universitária de Rio Grande (Rio Grande/RS). Patients enrolled between 2015 and 2017 (*n* = 264) were selected from those attending the outpatient clinic of the Endocrinology Service or Ophthalmology Service of the HCPA.

Type 2 DM was defined according to the American Diabetes Association guidelines [32]. The inclusion criteria were as follows: age ≥30 years at the time of T2DM diagnosis, no previous episodes of ketoacidosis, and no need to use insulin to treat diabetes within a period of one year after their diagnosis. As detailed elsewhere [33], all patients underwent physical examination, routine biochemical tests, and an interview to collect their demographic and clinical data, including the duration of their diabetes, smoking habits, and medication use. Clinical data were also obtained from medical records. The eGFR was calculated using the CKD-EPI formula [34], and diabetic kidney disease was defined based on the eGFR and albuminuria, according to the KDIGO 2022 guidelines [35]. After pupil dilation, DR was assessed by a retinal ophthalmologist at each participating institution using either ophthalmoscopy (for patients enrolled from 2000 to 2010) or retinography (for those enrolled from 2015 to 2017).

Blood donors were included in the study as a reference sample to determine the distribution frequency of the rs9896052 polymorphism in the general population. They were enrolled in the Hemotherapy Service of the HCPA between 2017 and 2018. Blood donors were interviewed to collect demographic and basic clinical data such as their medication use and family history of diabetes. Skin color was self-reported and categorized as white or non-white (brown/pardo or black). At the time of the enrollment interview, an 8 mL sample of peripheral venous blood was collected from each patient with T2DM and blood donors for molecular analyses.

### 4.3. Genotyping

DNA was isolated from the blood leukocytes using a standard salting-out method [36]. Genotypes of the rs9896052 polymorphism were identified via a real-time polymerase chain reaction (PCR) using a fluorogenic 5′ nuclease assay (TaqMan^®^, assay ID: C__30475753_10; Thermo Fisher Scientific Inc., Waltham, MA, USA). Eight microliter amplification reactions containing 10 ng of genomic DNA, 1X genotyping assay, and 1X TaqMan^TM^ Genotyping Master Mix were loaded into a real-time PCR thermal cycler (StepOnePlus Real-Time PCR System; Thermo Fisher Scientific Inc., Waltham, MA, USA) and subjected to the cycling conditions recommended by the manufacturer. Genotypes were scored independently by two blinded investigators (C.M.C.B. and L.M.d.S.M.) by inspecting the amplification plots using SDS software version 2.3 (Thermo Fisher Scientific Inc., Waltham, MA, USA). Positive and negative controls were used in each genotyping run; ambiguous samples were re-genotyped, and 104 samples (12.4% of the total) were randomly selected for amplification twice to check the accuracy of the genotyping. Two samples were not amplified and the concordance rate of the other samples was 100%. The genotyping data generated in this study are openly available in the FigShare repository [37].

### 4.4. Statistical Analysis

Categorical data were expressed as a percentage or absolute frequency (percentage), while quantitative data were expressed as a mean ± standard deviation or median (25th and 75th percentiles). Differences in the frequencies of categorical variables were compared between the groups using the chi-square test, with Yates correction and an analysis of adjusted residuals where appropriate. The chi-square test was also used to assess whether the genotypes were in Hardy–Weinberg equilibrium. After examining the normality of their distribution using the Shapiro–Wilk test, continuous variables were compared using the unpaired Student’s *t*-test, a one-way analysis of variance (ANOVA) with Tukey’s post hoc test, or their non-parametric analogs (Mann–Whitney U-test and Kruskal–Wallis followed by Dunn post hoc test), as appropriate.

Logistic regression analysis was used to estimate the odds ratio (OR) and the corresponding 95% confidence interval (95% CI) of the association between the rs9896052 polymorphism and PDR. The models were adjusted for skin color and covariates that were associated with PDR in the univariate analysis using the backward stepwise procedure. Analyses of the distribution of the rs9896052 polymorphism between the study groups were also stratified by skin color (white or non-white) and the known duration of diabetes (<10 years or ≥10 years). Statistical analyses were conducted using SPSS version 18 (SPSS Inc., Chicago, IL, USA) and WinPEPI version 11.43 [38]. *p* values < 0.05 were considered statistically significant.

Our sample size calculation indicated that at least 470 patients with T2DM (235 cases and 235 controls) would be needed to detect an OR of 1.45 (95% CI: 1.03–2.18) for the association between the A allele and PDR under the allele model, as observed by Burdon et al. [10] in their primary replication cohort of Australians with T2DM. For this calculation, we used an alpha error of 5% and a global frequency of 53% for the putative risk allele (A), as reported in the large TOPMED database [39]. The sample size was estimated using WinPEPI software.

## 5. Conclusions

In conclusion, the C allele of the rs9896052 polymorphism upstream of *GRB2* was more frequent in PDR cases than in controls, and the CC genotype was independently associated with an increased risk of PDR in Southern Brazilians with a known T2DM duration of less than 10 years. These findings support the hypothesis raised in a previous GWAS that the rs9896052 polymorphism is associated with severe forms of DR. Further studies in humans are needed to confirm whether and which of the alleles (A or C) could be useful for identifying patients at high risk of an early progression to PDR. In addition, functional studies are warranted to elucidate the mechanistic link between this gene variant, its expression, and PDR.

## Figures and Tables

**Table 1 ijms-25-10232-t001:** Clinical, laboratory, and demographic characteristics of patients with T2DM with PDR (cases) and without DR (controls).

Variable	Controls (*n* = 554)	Cases (*n* = 284)	*p* Value
Age (years)	58.9 ± 9.7	62.1 ± 8.5	<0.001
Male gender	208 (37.5%)	167 (58.8%)	<0.001
White	429 (77.4%)	222 (78.2%)	0.878
Age at T2DM diagnosis (years)	47.3 ± 9.4	45.8 ± 9.3	0.037
T2DM duration (years)	11.5 ± 7.2	16.3 ± 8.5	<0.001
Glycated hemoglobin (%)	7.7 ± 2.2	7.3 ± 1.8	0.134
Daily insulin use	38.0%	63.3%	<0.001
Body mass index (kg/m^2^)	30.6 ± 6.2	28.1 ± 4.8	<0.001
Smoking			0.017
Never smoker	51.6%	58.2%	
Former smoker	33.6%	33.8%	
Active smoker	14.8%	8.0%	
Hypertension	73.9%	79.2%	0.128
Systolic blood pressure (mmHg)	140.5 ± 22.6	144.7 ± 24.5	0.042
Diastolic blood pressure (mmHg)	84.1 ± 13.7	84.4 ± 12.6	0.788
eGFR (mL/min/1.73 m^2^)	97 (77–108)	60 (12–93)	<0.001
Diabetic kidney disease	50.0%	82.7%	<0.001
Total cholesterol (mmol/L)	5.07 ± 1.24	5.18 ± 1.39	0.456
HDL cholesterol (mmol/L)	1.11 (0.93–1.34)	1.02 (0.85–1.29)	0.009
LDL cholesterol (mmol/L)	2.87 (2.28–3.84)	3.05 (2.25–3.97)	0.417
Triglycerides (mmol/L)	1.78 (1.28–2.71)	1.78 (1.29–2.70)	0.743

Abbreviations: T2DM—type 2 diabetes mellitus; PDR—proliferative diabetic retinopathy; DR—diabetic retinopathy; eGFR—estimated glomerular filtration rate. Data are expressed as mean ± standard deviation, median (25th–75th percentiles), number of individuals (percentage), or percentage.

**Table 2 ijms-25-10232-t002:** Genotype and allele distribution of rs9896052 polymorphism in blood donors and patients with T2DM with PDR (cases) and without DR (controls).

Polymorphism	Blood Donors (*n* = 106)	Patients with T2DM				
		All Patients (*n* = 838)	*p* Value *	Controls (*n* = 554)	Cases (*n* = 284)	*p* Value **
Genotype						
AA	24 (22.6%)	188 (22.4%)	0.996	125 (22.6%)	63 (22.2%)	0.570
AC	52 (49.1%)	415 (49.6%)		280 (50.5%)	135 (47.5%)	
CC	30 (28.3%)	235 (28.0%)		149 (26.9%)	86 (30.3%)	
Allele						
A	100 (47.2%)	791 (47.2%)	>0.999	530 (47.8%)	261 (46.0%)	0.497
C	112 (52.8%)	885 (52.8%)		578 (52.2%)	307 (54.0%)	

Abbreviations: T2DM—type 2 diabetes mellitus; PDR—proliferative diabetic retinopathy; DR—diabetic retinopathy. *—*p* value for the comparison between blood donors and patients with T2DM (all); **—*p* value for comparison between cases and controls.

**Table 3 ijms-25-10232-t003:** Genotype and allele distribution of rs9896052 polymorphism in blood donors and patients with T2DM with PDR (cases) and without DR (controls), stratified by skin color.

Skin Color	Polymorphism	Blood Donors	Patients with T2DM				
			All Patients	*p* Value *	Controls	Cases	*p* Value **
White	Genotype	*n* = 83	*n* = 651		*n* = 429	*n* = 222	
	AA	15 (18.1%)	111 (17.1%)	0.928	73 (17.0%)	38 (17.1%)	0.206
	AC	41 (49.4%)	336 (51.6%)		231 (53.9%)	105 (47.3%)	
	CC	27 (32.5%)	204 (31.3%)		125 (29.1%)	79 (35.6%)	
	Allele						
	A	71 (42.8%)	558 (42.9%)	>0.999	377 (43.9%)	181 (40.8%)	0.299
	C	95 (57.2%)	744 (57.1%)		481 (56.1%)	263 (59.2%)	
Non-White	Genotype	*n* = 23	*n* = 187		*n* = 125	*n* = 62	
	AA	9 (39.1%)	77 (41.2%)	0.849	52 (41.6%)	25 (40.3%)	0.300
	AC	11 (47.9%)	79 (42.2%)		49 (39.2%)	30 (48.4%)	
	CC	3 (13.0%)	31 (16.6%)		24 (19.2%)	7 (11.3%)	
	Allele						
	A	29 (63.0%)	233 (62.3%)	>0.999	153 (61.2%)	80 (64.5%)	0.610
	C	17 (37.0%)	141 (37.7%)		97 (38.8%)	44 (35.5%)	

Abbreviations: T2DM—type 2 diabetes mellitus; PDR—proliferative diabetic retinopathy; DR—diabetic retinopathy. *—*p* value for the comparison between blood donors and patients with T2DM (all); **—*p* value for comparison between cases and controls.

**Table 4 ijms-25-10232-t004:** Genotype and allele distribution of rs9896052 polymorphism in patients with T2DM with PDR (cases) and without DR (controls), stratified by duration of diabetes.

T2DM Duration	Polymorphism	All Patients	Controls	Cases	*p* Value *
<10 years	Genotype	*n* = 281	*n* = 221	*n* = 60	0.052
	AA	62 (22.1%)	52 (23.5%)	10 (16.7%)	
	AC	133 (47.3%)	109 (49.4%)	24 (40.0%)	
	CC	86 (30.6%)	60 (27.1%)	26 (43.3%)	
	Allele				0.032
	A	257 (45.7%)	213 (48.2%)	44 (36.7%)	
	C	305 (54.3%)	229 (51.8%)	76 (63.3%)	
≥10 years	Genotype	*n* = 557	*n* = 333	*n* = 224	0.876
	AA	126 (22.6%)	73 (21.9%)	53 (23.7%)	
	AC	282 (50.6%)	171 (51.4%)	111 (49.5%)	
	CC	149 (26.8%)	89 (26.7%)	60 (26.8%)	
	Allele				0.831
	A	534 (47.9%)	317 (47.6%)	217 (48.4%)	
	C	580 (52.1%)	349 (52.4%)	231 (51.6%)	

Abbreviations: T2DM—type 2 diabetes mellitus; PDR—proliferative diabetic retinopathy; DR—diabetic retinopathy. *—*p* value for comparison between cases and controls.

**Table 5 ijms-25-10232-t005:** Univariate and backward stepwise multiple logistic regression analysis of the association between rs9896052 polymorphism and PDR in those with a short duration of T2DM (<10 years).

Variable	Unadjusted OR (95% CI)	Adjusted OR (95% CI)
CC genotype *	2.05 (1.14–3.70)	2.82 (1.17–6.75)
Male gender	3.89 (2.13–7.13)	6.24 (2.48–15.72)
Age (years)	1.04 (1.01–1.07)	1.06 (1.01–1.10)
HDL cholesterol (mmol/L)	0.13 (0.04–0.46)	0.06 (0.01–0.33)
LDL cholesterol (mmol/L)	1.72 (1.24–2.38)	2.29 (1.49–3.51)
eGFR (mL/min/1.73 m^2^)	0.98 (0.97–0.99)	0.99 (0.97–1.02) **
Non-white skin color	0.83 (0.41–1.69)	0.68 (0.11–4.06) ***
Glycated hemoglobin (%)	0.76 (0.62–0.93)	0.87 (0.64–1.17) ****
Body mass index (kg/m^2^)	0.85 (0.79–0.92)	0.94 (0.86–1.03) *****

Abbreviations: PDR—proliferative diabetic retinopathy; T2DM—type 2 diabetes mellitus; OR—odds ratio; CI—confidence interval; eGFR—estimated glomerular filtration rate. *—versus AC + AA genotypes; **—removed in step 2; ***—removed in step 3; ****—removed in step 4; *****—removed in step 5.

## Data Availability

The genotyping data generated in this study for patients with T2DM, along with a minimal demographic and clinical dataset, are openly available on FigShare (https://doi.org/10.6084/m9.figshare.26195513).

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
