# Peer review of "Association of the rs9896052 Polymorphism Upstream of GRB2 with Proliferative Diabetic Retinopathy in Patients with Less than 10 Years of Diabetes"

_ijms, 2024, doi:10.3390/ijms251910232_

Round 1
Reviewer 1 Report
Comments and Suggestions for Authors
The authors in this study investigated in patients from Brazil the relationship between rs9896052 polymorphism (A>C) located upstream of GRB2 and proliferative diabetic retinopathy (PDR with T2DM). The findings are interesting, and the articles make an significant impact to the metabolic disorders field.
The article is clearly presented, and discussion highlighting associations observed would have potential impact in the future. I would recommend the article for publication in the journal. I hope author would address any minor revisions suggested.
The researchers in this article addressed the specific polymorphism upstream of the GRB2 gene and its association with PDR in patients. The questions addressed are critical as they give information on potential markers to identify people at high risk for PDR, and this would lead us to enable earlier detection and apply targeted interventions.
The research article is identified as original and relevant to the field of PDR. The researchers addressed some specific gaps, primarily focused on the targeted population, which is generally underrepresented in genetic-related diseases and specific to DR. The knowledge generated through this article contributes valuable data to highlight the potential of population-specific differences involved in PDR.
The following are specific comments that could be improved for minor revisions.
For Table 5, including a brief description of clearer insight into the method or model selection in the table description would be a great improvement for clear understanding.
Comments on the Quality of English Language
Reviewer 2 Report
Comments and Suggestions for Authors
This study examines the correlation between the rs9896052 polymorphism, which is situated upstream of the GRB2 gene, and the occurrence of proliferative diabetic retinopathy (PDR) in patients from Southern Brazil who have type 2 diabetes mellitus (T2DM) and have been diagnosed with the disease for less than 10 years. The objective of the research is to ascertain the prevalence of the rs9896052 polymorphism in patients with PDR in comparison to controls without diabetic retinopathy. This analysis takes into account characteristics such as self-reported skin color and duration of diabetes.
1. How does the length of diabetes affect the genetic risk factors for developing PDR, specifically in patients with fewer than 10 years of diabetes?
2. How did you account for potential confounding factors, such as age, sex, glycemic control, and other diabetes comorbidities, that may have an impact on the occurrence of PDR?
3. Is it possible that there are genetic variables particular to certain populations that contribute to the observed differences in allele frequency distributions among the ethnic groupings in your study?
4. How can you resolve the absence of replication of this polymorphism's correlation with PDR in other groups or studies?
5. Is there functional evidence supporting the involvement of GRB2 in the development of diabetic retinopathy, which could explain the observed association?
6. The study reveals a notable correlation between the CC genotype and PDR in people who have had diabetes for less than 10 years. How does the molecular mechanism of GRB2 and its impact on insulin signaling and angiogenesis especially contribute to the initial stages of PDR development?
7. Given the genetic diversity within the population, how could genetic mixing affect association studies? Have you taken into account the use of admixture mapping or controlling for ancestry informative markers?
8. The study employs self-reported skin color as a substitute for ethnic heritage. What impact could misclassifying ethnic groupings have on the outcomes, and will using genetic ancestry analysis lead to more precise categorization?
9. Given the cross-sectional nature of the study, how can causality be established between the rs9896052 polymorphism and PDR?
10. What measures were implemented to guarantee that the non-diabetic control group accurately reflects the overall population, particularly considering the possibility of selection bias while recruiting blood donors?
11. What is the impact of environmental factors in Southern Brazil on the prevalence of the rs9896052 polymorphism and its connection with PDR, taking into account food trends, socioeconomic level, and healthcare accessibility?
12. The research examines various subgroup analyses without acknowledging the need for changes to account for repeated comparisons, which may heighten the likelihood of type I errors. It is crucial, especially when working with several layers and results.
Round 2
Reviewer 2 Report
Comments and Suggestions for Authors
The authors have replied to all my questions and comments satisfactorily.